



# Field-based landslide susceptibility assessment in a data-scarce environment: the populated areas of the Rwenzori Mountains

Liesbet Jacobs[1,2], Olivier Dewitte[1], Jean Poesen[3], John Sekajugo[4], Adriano Nobile[1], Mauro Rossi[5], Wim Thiery[6,7], Matthieu Kervyn[2]

[1] Royal Museum for Central Africa, Department of Earth Sciences, Leuvensesteenweg 13, 3080 Tervuren, Belgium
[2] Vrije Universiteit Brussel, Department of Geography, Earth System Science, Pleinlaan 2, 1050 Brussels, Belgium
[3] KU Leuven, Division of Geography and Tourism, Celestijnenlaan 200E, 3001 Leuven, Belgium
[4] Busitema University, Department of Natural Resource Economics, P. O. Box 236, Tororo, Uganda
[5] CNR-IRPI, Geomorphology division, via Madonna Alta 126, 06128 Perugia, Italy
[6] ETH Zurich, Institute for Atmospheric and Climate Science, Universitaetstrasse 16, 8092 Zurich, Switzerland
[7] Vrije Universiteit Brussel, Department of Hydrology and Hydraulic Engineering, Pleinlaan 2, 1050 Brussels, Belgium

*Correspondence to*: Liesbet Jacobs (liesbet.jacobs@vub.be)

**Abstract.** The inhabited zone of the Ugandan Rwenzori Mountains is affected by landslides, frequently causing loss of life, damage to infrastructure and loss of livelihood. This area of ca. 1,230 km$^2$ is characterized by contrasting geomorphologic, climatic and lithological patterns resulting in different landslide types. In this study, we focus on modelling the spatial pattern of landslide susceptibility based on an extensive field inventory constructed for five representative areas within the region (153 km$^2$) and containing over 450 landslides. To achieve a reliable susceptibility assessment, we investigate the effects of (1) using different topographic data sources and spatial resolutions and (2) changing the scale of assessment by comparing local and regional susceptibility models, on the susceptibility model performances using a pixel-based logistic regression approach. Topographic data is extracted from different the digital elevation models (DEMs) based on radar interferometry (SRTM and TanDEM-X) and optical stereo-photogrammetry (ASTER DEM). Susceptibility models using the radar-based DEMs generally outperform the ones using the ASTER DEM. The model spatial resolution is varied between 10, 20, 30 and 90 m. The optimal resolution depends on the location of the investigated area within the region but the lowest model resolution (90 m) rarely yields the best model performances while the highest model resolution (10m) never results in significant increases in performance compared to the 20 m resolution. Models built for the local case studies generally have similar or better performances than the regional model and better reflect site-specific controlling factors. On the regional level we investigate the effect of distinguishing landslide types between shallow and deep-seated landslides. The separation of landslide types allows to improve model performances for the prediction of deep-seated landslides and to better understand factors influencing the occurrence of shallow landslides such as topographic wetness, tangent curvature and total rainfall depth. Finally, the landslide susceptibility assessment is overlaid with a population density map in order to identify potential landslide risk hotspots, which could direct research and policy action towards reduced landslide risk in this under-researched, landslide-prone region.



# 1 Introduction

Landslide susceptibility assessments aim to estimate the probability of spatial occurrence of landslides given a set of geo-environmental conditions (Guzzetti et al., 2006). Susceptibility maps are fundamental tools for landslide hazard management, assisting governments, scientists or other stakeholders in policy decisions (Fressard et al., 2014). The methods used to achieve

these assessments are either knowledge-driven, process-based or statistical in nature (Guzzetti et al., 1999). Because statistical, data-driven models provide a quantitative assessment with reasonable data demands, these models are frequently applied at local to regional scales (Corominas et al., 2014). However, while literature is constantly growing with new susceptibility assessment techniques with increasing complexity (Korup and Stolle, 2014), some issues remain unresolved. Firstly, although statistical landslide susceptibility models are often applied, ambiguity regarding methodological issues on, among others,

landslide sampling, applied resolution and model uncertainty remains. Secondly, due to increasing computing capacities and the availability of dedicated software tools, landslide susceptibility models are sometimes applied without interpretation of the geomorphologic plausibility of the results (Steger et al., 2016). In addition, for many remote regions, reliable landslide susceptibility models are not available, despite the potentially large impacts of landslides on the local population in these areas. For example, for the African continent, a systematic under-investigation of landslides in all their facets is particularly

pronounced (Maes et al., 2017). Moreover, while recently, landslide susceptibility maps for these regions are issued at various scales ranging from continental (e.g. Stanley and Kirschbaum, 2017) to country-specific (e.g. Redshaw et al, 2017) and local (e.g. Che et al., 2012), the lack of regional and local assessments remains a major limitation in the identification and implementation of effective policy measures (Kervyn et al., 2015).

The Rwenzori Mountains in East-Africa are one of those regions where landslides pose a threat to life, livelihood and

infrastructure (Jacobs et al., 2016a; 2016b; Mertens et al., 2016). Although this threat is now generally recognized, thus far no quantitative landslide susceptibility assessment is available for the region. Additionally, given the potential role that landslides play in multi-hazard events (Jacobs et al., 2016c), an improved understanding of where landslides are likely to occur is needed. The primary objective of this work is to identify the factors conditioning the spatial occurrence of landslide events and as such better understand the patterns of landslide susceptibility at regional scale. Studies comparing different susceptibility modelling

approaches are frequent and often indicate minor differences between model performances (Dewitte et al., 2010; Zêzere et al., 2017). This research focuses on providing a reliable susceptibility assessment using a logistic regression modelling approach. This requires several methodological choices which will be explored either by consulting previous research findings available in scientific literature, for example with regard to landslide sampling or model construction, or by exploring through simulations the effect of using different sets of topographic data, altering spatial resolution, applying models at different scales

and separating landslide types. Therefore, this study aims to also fulfil a methodological objective by investigating the extent to which these factors influence the susceptibility assessment. Based on a critical geomorphological interpretation of these results, the final goal of providing a comprehensive landslide susceptibility assessment for the inhabited region of the Rwenzori





Mountain is pursued. This assessment is used to identify which uncertainties regarding landslides' controlling factors remain to be addressed and where potential hotspots for landslide risk may exist.

## 2 Study area and landslide inventory

The Rwenzori Mountains lie on the border of DR Congo and Uganda (Fig. 1). They cover an area of ca. 3,000 km² and reach an altitude of 5,109 m a.s.l.. In both countries, the flanks of the mountain are covered by natural vegetation which starts from 1,500-2,000 m a.s.l. and which is largely protected by a pristine national park (Fig. 1). The area of interest for this study is the densely inhabited zone below the national park borders in Uganda. This zone is relatively accessible for field surveys allowing the construction of landslide inventories. To map landslides in these areas, field surveys are required because of the limited possibilities to inventory landslides through remote sensing data interpretation (Jacobs et al., 2016b). This limitation is due to the very rapid vegetation recolonization of landslide areas or their reclamation by agriculture combined with a restricted availability of very high resolution optical imagery due to the persistent cloud cover in the humid tropics (Jacobs et al., 2016b). The Congolese footslopes of the Rwenzori are less populated and are more difficult to access. Moreover, for the Congolese side of the mountain no reports of landslides were found (Jacobs et al, 2016a).

The populated footslopes of the Ugandan Rwenzori Mountains, indicated on Fig. 1, cover ca. 1,230 km², including the lowlands in the north-west of the mountain range, where landslide densities are among the highest measured in the region (Jacobs et al., 2016b). Because of the large extent of the area of interest, targeted field surveys were conducted in 2014 and 2016, resulting in a landslide inventory for 5 study areas (153 km²), here referred to as Bundibugyo, Nyahuka, Kabonero, Mahango and Kyondo (Fig. 1). Together they represent all major lithological groups of the study region (Table 1). Kabonero, Bundibugyo and Mahango were inventorized in 2014. The landslide types in these three areas are described in Jacobs et al. (2016b). Nyahuka and Kyondo are two additional areas inventorized in 2016. For these two study areas a detail of the landslide inventory is given in Fig. 1. The sliding mechanisms in Kyondo are very similar to those in Mahango and Kabonero where mostly shallow soil and debris slides (estimated sliding plane <3 m deep) occur. Nyahuka has similar geomorphological characteristics as Bundibugyo, having a lowland section (with a hilly topography at elevations <1,000 m a.s.l.) and highlands characterized by metamorphic rock. The lowlands in the Bundibugyo study area are dominated by deep-seated rotational soil slides occurring preferentially in thick deposits of rift alluvium. In Nyahuka however, mostly shallow landslides prevail. A summary of the study areas surveyed, their areal extension, numbers of shallow and deep-seated landslides, their main lithologies and their average annual precipitation depth are given in Table 1. Rainfall is the main trigger of landslides in the region with only 14 landslides reported to be earthquake-triggered. In total this inventory contains 454 landslides used for the susceptibility modelling. More details on the landslide inventory construction can be found in Jacobs et al. (2016b).





## 3 Methodology

We apply a pixel-based logistic regression model to assess landslide susceptibility of the study area using landslide occurrence derived from the landslide inventory as dependent variable. The logistic regression model is a widely applied statistical approach for predicting dichotomous dependent variables, such as the presence or absence of landslides (Hosmer and
Lemeshow, 2004; Brenning, 2005):

$$P(y = 1) = \frac{1}{(1+\exp-(\alpha+\beta1X1+\beta2X2+\cdots+\beta nXn))} \quad (1)$$

With *y* the dichotomous variable indicating the presence or absence of a landslides, *Xi* the explanatory variables considered by the model and βi the coefficients assigned to each explanatory variable *Xi*. The output probability values range from 0 to 1,
corresponding to a zero to 100% spatial probability of a landslide occurrence. The logistic regression has the advantage of enabling a straightforward interpretation of which independent variables contribute to the prediction and how they do so (Tu, 1996).

When applying a pixel-based statistical landslide susceptibility model such as the logistic regression model, several methodological choices need to precede the construction of the model and the final susceptibility assessment. These
methodological choices can notably include the scale of the assessment, the model's spatial resolution (Tian et al., 2008), the subdivision between model calibration and validation data (Hussin et al. 2016) and the landslides sampling strategy (Nefeslioglu et al., 2008). The motivations behind these methodological choices are not always given, limiting the reproducibility of the analyses and the extent to which obtained results can be critically evaluated.

In the following sections we explicate and motivate the methodological choices made for our specific case study. Where
possible we base ourselves on recommendations found in key methodological research and available knowledge on landslide processes in the Rwenzori region. These choices are summarized in Fig. 2 and further explored below. For choices that depend on the characteristics of the case study, such as the applied scale and spatial resolution of the assessment, the topographic data source used, or the landslide type subdivision, several scenarios are simulated and the outcomes evaluated. This evaluation considers jointly (1) the performance and stability of the resulting model and (2) the degree to which the resulting model can
be interpreted and reflects a geomorphologic reality. First, choices regarding the applied spatial scale and resolution and related topographic data source are explored. Afterwards, the model construction itself is further discussed with regard to the landslide sampling strategy, the landslide type subdivision, the model variable selection and the model calibration and validation. Subsequently, the model evaluation method is specified. Finally, we discuss the assessment of landslide risk hotspots based on the regional landslide susceptibility map.

### 3.1 Selection of spatial resolution, topographic data source, and spatial scale of the analysis

In studies using pixel-based landslide susceptibility models, the choice of model's spatial resolution is often motivated based on the resolution of the available datasets, with a preference for using the most finely gridded resolution possible. In the context



of statistical landslide susceptibility assessments, only few studies compared different model resolutions (e.g. Tian et al., 2008). Additionally, because of the availability of multiple global digital elevation models (DEMs) such as the Shuttle Radar Topography Mission (SRTM DEM: NASA JPL., 2013), DEMs derived from optical ASTER imaging (ASTER DEM: METI/NASA, 2009) and often also DEMs derived from local topographic data, more than one source of topographic

information is available for landslide studies. Despite this, these various topographic information sources are rarely compared (e.g. Havenith et al., 2006). In the data-scarce setting of the Rwenzori Mountains, the selection of the most appropriate model resolution and topographic data source is particularly relevant. Here, to model landslide susceptibility, three different topographic information sources are used at four different resolutions.

The SRTM and ASTER DEMs are freely available online for most parts of the world. They are downloaded at the provided 1

arc-second (~30 m) resolution (METI/NASA, 2009; NASA JPL., 2013). The third topographic source is a TanDEM-X DEM. We constructed this DEM at a ~5 m resolution with the InSAR technique (Interferometric Synthetic Aperture Radar - Bürgmann et al., 2000; Hanssen, 2001) using TanDEM-X bi-static images (Moreira et al., 2004) acquired in ascending and descending orbit. For more background on the TanDEM-X DEM construction, we refer to Appendix A. The final TanDEM-X DEM is resampled to a 10 m resolution with subsequent aggregation to 20 m and 30 m resolution using weighted aggregation

(Grohmann, 2015). The ASTER DEM and SRTM DEM 1-second are resampled to precisely 30 by 30 m DEMs using bi-cubic resampling (Metz et al., 2010). Similar to the TanDEM-X, the SRTM30 is up-scaled to 90 m using weighted aggregation resampling. The variants based on TanDEM-X DEMs are hereafter referred to as TANDEMX10, 20 and 30, those based on SRTM as SRTM30 and 90 and those on ASTER DEM as ASTER30. While the TanDEM-X DEM and the SRTM are based on InSAR data, the ASTER DEM is produced by stereo-photogrammetry principles. By testing all three DEMs at the 30 m

pixel size, the DEM suitability regardless of the pixel size can be assessed. In total thus six different combinations of DEM sources and resolution are tested. Henceforth they are referred to as model variants.

In this study we also assess the dependency of the model on the spatial scale of the assessment. This is achieved by building models at the local level for individual case studies (ca. 20-43 km²) and at the regional level for all case studies combined (153 km²). Mahango and Kyondo are investigated together at the local scale due to their proximity and given the low number of

landslides in Kyondo. Kabonero, Bundibugyo and Nyahuka are investigated separately at the local scale. These four local scales and the regional scale are hereafter referred to as levels. The model combining all case study areas thus makes use of the landslide data of all case studies, which combined are considered to be representative for the Rwenzori inhabited zone. It should be noted that the calibration and validation are performed within the surveyed area boundaries and that outside the surveyed areas the susceptibility assessment should be considered as an extrapolation of the statistical model.





### 3.2 Model construction

#### 3.2.1 Landslide sampling

The sampling of landslides for pixel-based susceptibility modelling is not a well-defined procedure, and different approaches exist to select those pixels best representing the conditions under which the landslide occurred. While some studies take the

whole polygon defining the landslide boundaries as input, others consider either the centroid, a portion of the highest pixels within the landslide, the source area or construct a buffer zone around (portions of) the landslide to represent the conditions under which the landslide occurred (Dai and Lee, 2003; Suzen and Doyuran, 2004; Van den Eeckhaut et al., 2006; Che et al., 2012; Hussin et al., 2016).

Here, the approximate location of the landslide depletion zones was identified in the field and is thus available in the inventory

(Jacobs et al., 2016b). Although selecting all pixels within the depletion zone generally provides better model fits (Hussin et al., 2016; Yilmaz, 2010), this is likely an artefact due to the introduction of very similar pixels within the model (Hussin et al., 2016). Additionally, it can induce spatial autocorrelation, violating the assumption of independent observations for general linear models. Although some models take this into account, they are reported to be numerically more demanding and less stable (Brenning, 2005). Therefore, in order to avoid spatial-autocorrelation and equally consider landslides regardless of their

size, only the centroid of the depletion zone is used to represent the event.

#### 3.2.2 Landslide type distinction

As described in Sect. 2, the Rwenzori Mountains host a diversity of landslide types. Because each landslide type is expected to be controlled by a different set of explanatory variables or a different effect of those variables, separating the types is a

meaningful procedure in landslide susceptibility assessment. However, by doing so, sample sizes are reduced and statistical significance can be lost. In this study we compare, on the regional level, the effect of separating landslide types between the deep-seated (depth of sliding plane > 3m) and shallow landslides (depth sliding plane < 3m). The depth of the sliding plane was estimated in the field (Jacobs et al., 2016b). The separation of shallow and deep-seated landslides at the regional level allows to maintain reasonable sample sizes.

#### 3.2.3 Variable selection

From previous exploratory work on landslide data collected in the region, topographic and lithological variables appear to have considerable effects on landslide types and spatial distributions (Jacobs et al., 2016b). The topographic information extracted from the three different DEMs described in Sect. 3.1 are slope, elevation, profile- and tangent curvature, the aspect

and the topographic wetness index (TWI) defined as the natural logarithm of the ratio of the specific upstream contributing area over the tangents of the slope (Beven and Kirkby, 1979). The TWI serves as a proxy for spatial soil moisture patterns in





the landscape (Yilmaz et al., 2010). The aspect is considered as the sine and cosine of both the aspect expressed as degrees counter clock of east to express maximum differences in the north-south and east-west axis respectively as well as the aspect in degrees counter clock of north-east to express maximum differences between north-west and south-east as well as north-east and south-west respectively (Chang et al., 2007; Stage and Salas, 2007). Information on the lithology is extracted from

the lithological map of Uganda at 1:100,000 (GTK Consortium, 2012; Table 1) and converted to dummy variables with gneiss used as the reference lithology.

Detailed land use mapping serving as input for the landslide susceptibility model is in this study not feasible due to the complexity (multiple cropping and multi-layered) and dynamics of the land use with regard to agricultural crops (rapid alterations due to a bimodal rainfall pattern). However, especially for shallow landslides, a stabilizing effect of trees on the

soil could be expected. Therefore, at the regional level where shallow and deep-seated landslides are separated, the tree-cover percentage as calculated by Hansen et al. (2013) for the baseline year of 2000 is introduced into the model for shallow landslides. Additionally, for all regional simulations, annual average precipitation data on a 7*7 km² grid obtained by the COSMO-CLM² regional climate model, covering a period from 1998-2008, are used to analyse the effect of spatial precipitation distribution on landslide susceptibility (Thiery et al., 2015). Due to the relatively large pixel-size of this data set

compared to the area of the case studies, the risk exists that single pixels in the precipitation data set completely control precipitation amounts in the case studies. To avoid this, we apply a 3-by-3 averaging filter on the precipitation dataset, thereby ensuring a reliable representation of precipitation patterns on the regional level. Finally, although elevation is commonly used in landslide susceptibility models, its value in our regional model can be questioned. Elevation is often introduced as a proxy for rainfall depth and/or weathering (Coe et al., 2004; Dai et al., 2003), with the assumption that higher elevations are linked

to more rock weathering, leading to a weaker lithology. However, in the Rwenzori, rainfall distribution on the regional level is strongly linked to prevailing climatic systems not represented by the elevation alone (Thiery et al., 2015; Jacobs et al., 2016a). Furthermore, precipitation is already introduced in the regional model as an explanatory variable. In addition to this, on the regional level, low elevation is expected to reflect mostly higher landslide densities observed on a rift alluvium lithology rather than serving as a proxy for spatial rainfall patterns, as higher elevations are linked to more resistant lithological groups

(Table 1, Jacobs et al., 2016b), which are already introduced into the model. Therefore, in the regional model assessments, no strong arguments support the use elevation as an explanatory variable. For completeness, the regional models are run both with and without elevation.

The final variable selection on all levels and variants is performed by a stepwise selection procedure applied to all variables except the categorical variable of lithology for which all dummies are considered together in the model (Agresti 2003;

Heumann et al., 2017). Forward, backward and both ways stepwise selections are performed and the selection procedure for the final model and its variables is based on a minimization of the Akaike information criterion (AIC) which penalizes models with a large number of parameters and models with poor fit (Goetz et al., 2011).





### 3.2.4 Model calibration and validation and implications for stability

Similarly to landslide sampling, different approaches exist for separating the landslide inventory into a calibration and validation dataset. A minimum of 10-20% of the landslide population reserved for calibration is reported by Hussin et al. (2016). In most cases, the training dataset is chosen to be equal or slightly bigger than the validation set with 50-50%, 75-25% or 60-40% as common subdivisions. Studies comparing model performances at different subdivisions tend to show that a minimum of 10-20% of the landslides for the calibration is required after which an increase does not significantly improve the modelling results (Hussin et al., 2016). Therefore, in this study, for each variant, the landslides in the inventory are split randomly in equal proportion datasets equally serving in the calibration and validation stage respectively. Because this random split could influence the model construction, especially in these case studies dealing with small landslide datasets, this procedure is repeated 20 times for each model variant on each level to verify its stability. Pixels where no landslide occur are considered to be all the pixels in the study areas outside the landslide polygons. An equal portion of landslide and non-landslide pixels are used in the model calibration stage (Brenning, 2005; Hussin et al., 2016). All other pixels are used in the validation stage.

### 3.3 Assessment of model performances and comparisons

The above methodology results in a set of models of which performances are influenced by the topographic data source used, the applied resolution and represented scale. To assess the effect of these factors, first the model variants are compared within each level, after which the regional level is compared to the local ones.

The evaluation of model performances using the different topographic data sources and various spatial resolutions at each level (model variants) is based on the comparison of the model fit in terms of $AUC_{ROC}$, a summary statistic indicating the model performance corresponding to the area under the receiver-operating characteristics curve which combines both sensitivity and specificity (Fawcett, 2006). $AUC_{ROC}$ values of 1 indicate a perfect model fit, while values of 0.5 indicate that the model is not performing better than a random distinction. Generally, models with $AUC_{ROC}$ values greater than 0.7 are considered to have an acceptable model performance (Fressard et al., 2014). The repeated random sampling of landslides for the calibration and validation of the models explained in sect. 3.2.4. results in a sample of 20 $AUC_{ROC}$-values per model variant. For each level, the performance differences between (1) all variants of DEMs and resolutions tested, (2) the three different tanDEM-X variants, (3) the two different SRTM variants and (4) the three different variants on the 30m resolution is assessed. The tests used for this depend on whether or not the assumption of normality and homoscedasticity can be made. Therefore, the distribution of $AUC_{ROC}$ values for all model variants is first tested for normality using the Shapiro-Francia test (Thode, 2002). Afterwards, homoscedasticity is tested for each comparison using the Levene test and the Fligner-Killeen test for normal and non-normal distributions respectively (Conover et al., 1981; Glass 1966). For comparisons of variants that are normally distributed and homoscedastic, the ANOVA single-factor tests are used to compare mean performances over more than two variants, while



Student's t-tests are used to evaluate the difference between mean performances of two variants (Lowry, 2014). For non-normal distributions that are homoscedastic, non-parametric tests comparing variant's sample medians are introduced to complement the parametric tests: the Kruskal-Wallis test to compare more than two model variants, and the Mann-Whitney-U test to directly compare two variants (Lowry, 2014). In case variants are found to be heteroscedastic, an additional Welch-test is performed to investigate the difference between variants (Welch, 1951). Based on these assessments, variants for each applied level (local and regional) will be selected. To further evaluate these selected variants' model predictive skill, sensitivity, specificity and prediction curves are considered separately (Guzzetti et al., 2006, Van Den Eeckhaut et al., 2009).

For the evaluation of the effect of applied scale on the susceptibility assessment, a pairwise comparison of the regional model to the different local models is performed visually and in terms of the variables selected within each model. To quantitatively assess the ability of the selected regional model to predict local landslide susceptibility, the regional model is additionally validated using the local dataset of landslides and non-landslides at each local level after removal of the points used for the regional model's calibration. Discrepancies surfacing through these comparisons will allow to identify potential strengths or limitations of the application of the regional model to the entire inhabited zone.

## 3.4 Regional landslide susceptibility assessment and preliminary identification of risk hotspots

Landslide susceptibility- and risk zonation are particularly rare in equatorial Africa but all the more relevant in focusing research and policy action (Kervyn et al., 2015). Here, we use regional landslide susceptibility and population density distribution to provide a preliminary identification of landslide risk hotspots. The regional susceptibility model for the Rwenzori Mountains is obtained by applying all 20 simulations of the optimal model variant and averaging the resulting susceptibility values. Subsequently, parish population density data for the year 2002 provided by the Uganda Bureau of Statistics (UBOS, 2003) are rescaled to continuous values between 0 and 1. By multiplying the rescaled population density with the regional landslide susceptibility, a first identification of potential landslide risk hotspots can be made. Evidently, the approach applied here is rough and among others does not account for within-parish population concentration, temporal aspects of the landslide occurrence nor vulnerability or resilience of the population to the occurrence of a landslide event. The goal is not to present a risk assessment but to explore where hotspots for landslide risk could possibly occur and consequently which regions require particular attention when considering landslide risk assessments.

## 4 Results



## 4.1 Influence of model's spatial resolution, topographic data source and applied scale

Variants' model performances for each level in terms of mean and standard deviation of the $AUC_{ROC}$ in the validation is given in Table 2. With regard to the regional model, no statistical difference between models with and without the introduction of elevation is found (paired t-test, $p<0.05$). Therefore, the regional model without elevation is used throughout this study. At the

regional level and all local levels, the null-hypothesis that all models variants have the same performance can be rejected (Table 2). When comparing the DEMs at 30 m at any level, ASTER30 never produces the best model results (Table 2). Similarly, the SRTM90 performs significantly worse than SRTM30 at the regional level as well as all local levels with the exception of Nyahuka where the SRTM90 outperforms the SRTM30 (Table 2). With regard to the optimal resolution of the TanDEM-X variants, there is only a significant difference between TanDEM-X variants in Nyahuka and Kabonero. The

increase in resolution from 20 to 10 m is tested separately and does not increase performances at any level ($p<0.05$).

Table 2 also indicates that the best performing local models with the exception of Bundibugyo have better performances than the best regional model in terms of $AUC_{ROC}$. At the local level, the best performing model variant depends on the location of the case study area but in most cases, no single model variant significantly outperforms another. Only in Nyahuka, one variant – the SRTM90 variant - clearly outperforms the others. In Kabonero and Kyondo/Mahango, the TanDEMX20 is preferred and

the SRTM30 provides the best results for Bundibugyo. For the local levels, the variant with the highest $AUC_{ROC}$ is selected and indicated in bold in Table 2. Because of the potential influence of the accuracy of the TanDEM-X DEM on the regional susceptibility assessment outside the studied zones, the consistent performance of the SRTM DEM and the lack of performance difference between the TANDEMX and the SRTM30 variants, the SRTM30 variant is selected as the basis for the regional landslide susceptibility map. For the regional and all local levels, the selected model variant are further explored in terms of

sensitivity, specificity and prediction rate (Fig. 3). The results show that also for these performance indices, generally, the local models outperform the regional model with the exception of model sensitivity in Nyahuka and Bundibugyo and model prediction rate in Bundibugyo. However, the variation on these model evaluators between simulations for the local models is also larger than that for the regional model.

Based on these results, the selected variants are interpreted in terms of their selected variables. Table 3 summarizes how many

times over the 20 simulations a certain explanatory variable is selected by the model variant indicated in bold in Table 2, which sign the corresponding β-value has, and how often these variables are considered to be significant in the logistic regression. Slope is a significant contributor in all study areas and in nearly all runs. Elevation plays different roles at the local levels with a positive influence on the occurrence of landslides within Kyondo/Mahango and a slightly negative influence in Bundibugyo. TWI has a positive response in the landslide susceptibility assessment in Nyahuka and to a lesser extent in Kyondo/Mahango,

while the opposite is observed in Kabonero. Aspect only seems of limited importance in the Bundibugyo where south-east and south-facing slopes appear more landslide-prone than north-west and north-facing slopes, respectively. Profile concave slopes favour landslides in Nyahuka while profile convex slopes favour landslides in Kabonero and Bundibugyo. Finally, tangent



concave slopes favour landsides on the regional level, in in the typical highland regions of Kabonero and -to lesser extent- of Kyondo/Mahango.

With regard to lithology, some classes have a more readily interpretable behaviour than others. Rift alluvium is almost invariably found to be positively indicative for the occurrence of landslides compared to gneiss. In most cases this effects are

also found to be significant. Amphibolite is strongly negatively associated to the occurrence of landslides compared to gneiss on the regional level and in Kabonero but positively associated to landslides in Nyahuka. Mica schists also have complex behaviour, with a negative influence on landslide presence in Kabonero and Kyondo/Mahango but a positive effect in Bundibugyo and Nyahuka compared to the reference lithology, gneiss. Finally quartzite is found to mostly favour landslides albeit non-significantly.

To evaluate the difference of the regional susceptibility model with the local models, the susceptibility maps are pair-wise displayed in Fig. 4. Overall, topographic patterns are consistent over the pairs, except for Nyahuka where the strong imprint of TWI in the local model is evident. Furthermore, important visual discrepancies are present along lithological boundaries in Nyahuka and Kyondo. The local model for Nyahuka appears to be less influenced by lithology compared to the regional model. In Kyondo the opposite can be observed, with more pronounced lithological effect on the local susceptibility model as to the

susceptibility assessment predicted by the regional model. This is in accordance with the findings from Table 3 where amphibolites are positively selected in the local Nyahuka model in contrast to the negative selection in the regional model, while mica schists are more often selected as a negative predictor in the local model for Kyondo than in the regional model, and therefore contrast more with gneiss and quartzite. These visual discrepancies are also translated in a lower $AUC_{ROC}$ of the regional SRTM30 model when validated at the local levels individually (Fig. 4), compared to the $AUC_{ROC}$ for the SRTM30

variants of the local levels (Table 2), with the exception of Kabonero. In other words, models calibrated and validated at the local level generally outperform the regional model applied to that level.

## 4.2 Separation of landslide types

Based on the findings above, the SRTM30 is applied for the separation of landslide types. The model fit for shallow landslides slightly decreases (average $AUC_{ROC}$ 0.67) compared to a model encompassing all landslides. For deep-seated landslides, the type separation improved the model performance with an increase of $AUC_{ROC}$ from 0.71 to 0.81. The variables selected by both models are summarized in Table 4. Major differences between the model for shallow and deep-seated landslides can be found in the response to lithology, topography and precipitation. With regard to lithology, rift alluvium is a significant

explanatory variable for deep-seated landslides but less often so for shallow landslides. In contrast, the presence of amphibolite is a strong negative predictor for shallow landslides but rarely significant for deep-seated landslides. With regard to topography, the positive influence of TWI appears in the model for shallow landslides but does not play a role in the model for deep-seated landslides. Likewise, the influence of tangent concave slopes becomes less important in the model for deep-seated landslides.





Furthermore, the effect of annual average precipitation disappears in the model for deep-seated landslides. Finally, the percentage tree-cover introduced in the model for shallow landslides is rarely selected and never found to be significant.

### 4.3 Regional landslide susceptibility and population distribution: identifying risk hotspots

The regional susceptibility model is obtained by applying all 20 model simulations of the regional SRTM30 variant for all

landslides and subsequent averaging (Fig. 5a). This approach seems to produce some artefacts in the north sections of the lowland region (red arrow, Fig. 5a). Here, even at low slope gradient, medium to high landslide susceptibility values are assigned because of the occurrence of rift alluvium, strongly positively connected to landslides in the regional model. However, it could be expected that slopes between 0-5˚ generally do not favour the occurrence of landslides in these lowlands (Jacobs et al., 2016b). Therefore pixels with a slope gradient less than 5˚ are reclassified. To avoid an abrupt change in susceptibility

values at this threshold, the susceptibility in these mapping units is not set to zero but rescaled to values between 0-0.35 (i.e. rescaled to the lowest susceptibility class) according to their initial susceptibility value (Fig. 5b). This corrected landslide susceptibility map is combined with the rescaled population densities provided by UBOS (2003) (Fig. 5c) to produce a preliminary identification of landslide risk hotspots (Fig. 5d).

### 5 Discussion

### 5.1 Effect of the model's spatial resolution and applied DEM on landslide susceptibility:

With regard to the spatial resolution selected, it has to be noted that an increase in model resolution does not necessarily influence the explanatory variables in equal ways: DEM resolution has potentially very different influences on the actual values

derived for elevation, slope or other derivatives (Dewitte et al. 2010, Kervyn et al. 2008). In this case, a decrease of spatial resolution from 30 to 90 m in general decreased model performances. An increase in resolution from 20 to 10 m however does not result in performance increases which is in accordance with the concept of "optimal model complexity" where an increase in information does not necessarily improves the model performances (Grayson et al., 2002) but instead performances tend to reach an optimum after which increasing data availability does not increase or even decreases performance (Dewitte et al.,

2010).

Besides spatial resolution, the DEM source also influences the predictive power of the applied susceptibility model. The DEM based on optical imagery (ASTER) at the 30 m resolution never results in susceptibility models that significantly outperform those based on InSAR technology (SRTM and TanDEM-X) and on four levels, the SRTM30 and TANDEMX30 significantly outperform the ASTER30 variants. This is supported by earlier comparisons of the SRTM and ASTER DEMs by Kervyn et





al. (2008) who found that the ASTER DEMs have lower vertical accuracies and that they are affected by more small-scale noise and resulting apparent topographic variability than the SRTM DEM.

## 5.2 Variables influencing landslide susceptibility

With the exception of Bundibugyo, all local models outperform the regional model in terms of $AUC_{ROC}$ and prediction rate. In general, by increasing the scale from regional to local level, more and more similar landslide processes are simulated within a single model. Because different landslide processes are controlled by different geo-environmental conditions, a downscaling to the local level can thus potentially result in a better performing model more tailored to the local conditions and sliding

processes. An adverse effect can be observed in Bundibugyo, characterized by two very diverse sets of mass movements with deep-seated rotational slides in its lowlands and shallow soil- and debris slides in its highlands (Jacobs et al. 2016b). Therefore, downscaling from the regional to the local level for this case study area does not increase the model performance. Additionally, it is important to point out that the landslide sampling procedure used here assumes that the topography at the location of the depletion centroid was not altered. Particularly in the lowlands in Bundibugyo where large landslides are likely to leave

important topographic signatures, this assumption could be violated, which could influence model performances. The increase in variation (or the decrease in stability) of $AUC_{ROC}$, sensitivity and specificity and prediction curves on the local levels compared to the regional simulations can be explained by a decrease in landslide sample size for models built on the local level.

The visual analysis of the resulting susceptibility maps and the interpretation of their selected variables, in general reveal

similar patterns between the local and the regional landslide susceptibility model. However, important exceptions to this can be found in Nyahuka where there is a strong imprint of TWI on the local susceptibility map and in Nyahuka and Kyondo where local and regional susceptibility differences are very pronounced for particular lithological groups. In Nyahuka, landslides in the lowland portion of the study area concentrate along rivers (Fig. 4). At the 90m resolution, pixels with high TWI are large enough to include the depletion zone of the centroid, explaining the strongly positive influence of TWI in the local model and

the exceptional preference for the SRTM90 variant. Additionally, in the regional model for Nyahuka, pixels on amphibolite have very low susceptibility values. This contrasts with the Nyahuka local model, where amphibolites are considered to favour the occurrence of landslides compared to gneiss. An adverse effect is noticeable in Kyondo, where mica schists are relatively less susceptible than what the regional model predicts. In Nyahuka, this unexpected lithological response of amphibolite could be due to the mapping accuracy of the lithological map. The lithological map used here is the most recent and most detailed

map available for the region. However, the producers of the map report that field observations are limited to a section along the eastern foot-hills (Geological Survey of Finland, GTK, 2014). Previous lithological maps found in literature do not show the presence of an amphibolite group in the Nyahuka region (Bauer et al., 2010; Koehn et al., 2010; GTK, 2014). The landslides located on the amphibolite lithology in Nyahuka occur close to the lithological boundaries (Fig. 4) and therefore relatively



small mapping errors could lie at the basis of this different apparent response of amphibolites in this region. For both Nyahuka and Kyondo the unexpected local lithological effects could also be due to regional differences between lithological groups not reflected in the classification system. It is possible that lithologies belonging to the same group will have different weathering patterns due to different climatic or tectonic regimes and thus different effects on landslide susceptibility. This is also stated

by Dewitte et al. (2010), who point out that the relevance of a variable in a landslide susceptibility assessment does not only depend on whether that variable plays a role in the landslide process itself but also depends on the quality of that variable. Finally, the local models in general outperform the regional model applied to the local levels (Fig. 4). This supports the observation of visual discrepancies between local and regional maps. In summary, the local maps are more tailored to represent the spatial probability of the sliding processes in that area and should therefore, wherever available, be preferred over the

regional model.

### 5.3 Separation of landslide types

The separation of landslide types influences the predictive powers of susceptibility models improving the performances deep-

seated landslides susceptibility zonation. This can be explained by the strong dependency of deep-seated landslides to lithology and in particular on rift alluvium, which is selected and found significant in all runs for each model considering only deep-seated slides. Rift alluvium in the Rwenzori Mountains is characterized by deep clay-rich deposits, lacking solid bedrock and therefore providing a medium for deep-seated shear planes.

Also for shallow landslides, the separation of landslide types allows a better understanding of their regional controls, despite

a decrease in model performance. Notably the consideration of only shallow landslides allows to better isolate topographic and climatic effects: the concentration of soil moisture in the landscape, represented by the TWI, tangent curvature and the annual average precipitation distribution are important positive predictors of shallow landslides at the regional scale while their effect weaken or disappear in the model for deep-seated landslides. Finally, tree cover fraction was expected to be a negative predictor for the occurrence of shallow landslides due to the potentially stabilizing effect of tree roots on the hillslopes, however this is

not supported by the model results. This, however, does not a priori indicate that tree cover is not important predictor but might relate to limitations of the datasets itself. Firstly, as the tree cover fraction is taken from the baseline year of 2000, it does not account for alterations preceding the most recent landslides in the inventory. Unfortunately, detailed and reliable temporal information on the mapped landslides is often not available, limiting the possibility to include land cover (change) as a reliable predictive variable. Secondly, in the tree cover fraction classification, confusion exists between tree cover and cocoa, a

common land use in the lowland areas of Bundibugyo and Nyahuka where nearly half of the shallow landslides occur, potentially resulting in an insensitivity of the regional model to tree cover fraction.





## 5.4 Regional landslide susceptibility and population distribution: identifying risk hotspots

From Fig. 5d, which shows where high landslide susceptibility co-occurs with a high population density, some zones can be isolated presenting apparent hotspots for landslide risk. First of all, in Bundibugyo town and its surrounding parishes,
population densities and landslide susceptibility are very high, leading to a potential hotspot (Fig. 5d: I.). However, also on the western flank of mountain's horst itself, a high landslide susceptibility is combined with high population densities (Fig. 5d: II.). This observation is in contrast to the north-east flanks of the Rwenzori mountains where higher landslide susceptibilities are mostly combined with low population densities or vice versa (Fig. 5d. zones A and B respectively). In the south-east of the Rwenzori mountains, the area around Kilembe, located in the Nyamwamba valley appears to be a potential hotspot for landslide
risk. Moreover, this valley is also known for its flash flood risk (Jacobs et al., 2016c). Currently, the construction of a hydropower station in the Kilembe valley is ongoing. To which extent landslide and flood risks are taken into account in its design is unclear. South of the towns of Kasese and Kilembe, high landslide susceptibility values are again often combined with high population densities (Fig. 5d: III. to V.). Hotspot III. largely coincides with Mahango Sub-County: a region where landslides have regularly caused fatalities. Hotspots IV. and V. are so far very little researched. These areas are not
systematically investigated, nor did they appear as a landslide hotspot in archive studies (Jacobs et al. 2016a). In an effort to collect more landslide data in various remote areas of the Rwenzori region, a network of 20 inhabitants, referred to as geo-observers, is established. The geo-observers are trained to report on natural hazards occurring within their environment, among which landslides, by means of a digital questionnaire filled in using smartphone devices (VLIRUOS, 2017).

One needs to stress that this approach for identifying potential landslide risk hotspots entails major constraints and results
should be considered as indicative and interpreted with extreme care. Evidently, the use of only landslide susceptibility and population density is a crude simplification of potential landslide risk. Moreover, the regional susceptibility model does not have a perfect performance and is, as stated above, an extrapolation for those areas outside the surveyed case studies. Landslide risk is also intrinsically connected to the size and speed of the landslide, parameters which are not accounted for here. Finally, within-parish distribution of population densities can significantly influence landslide risk, as within-parish spatial variation
of landslide hazards can be expected and population dynamics since 2002 are not taken into account. This assessment is therefore not an estimate of landslide risk, but should be considered as a first indication of where research and policy priorities should be concentrated.

## 6 Conclusions

In this study we aimed to provide a reliable landslide susceptibility assessment for the inhabited region of the Rwenzori
Mountains by analysing the effects of the considered scale (local assessments vs. regional assessment), topographic data sources and their spatial resolution as well as the separation of the landslide types. With regard to the DEM source,





susceptibility models based upon DEMs derived from InSAR products (SRTM and TanDEM-X) generally outperform the DEM derived from optical imagery (ASTER). While a resolution decrease of 30 to 90 m generally decreases model performances, an increase from 20 to 10 m does not improve model performances. The separation of landslide types at the regional level allows to improve model skills for deep-seated landslides and to better understand the factors contributing to the

susceptibility of shallow landslides. This study shows that at the regional level, slope and prevailing lithology strongly controls landslide susceptibility. Shallow landslides seem to be more controlled by regional rainfall distribution and local runoff concentration in the landscape while a strong effect of the presence of rift alluvium influences the occurrence of deep-seated landslides.

Recent research efforts have led to an increased availability of global, regional, and even country-specific landslide

susceptibility maps, also for data-scarce regions such as equatorial Africa (Broeckx et al., 2017; Redshaw et al., 2017; Stanley and Kirschbaum, 2017). In contrast, local and regional susceptibility assessments remain particularly rare in these regions. For the Rwenzori Mountains, we found that the local susceptibility assessments are generally better suited for representing the site specific controlling mechanisms of landslides. In parallel to smaller scale landslide susceptibility studies, adequate attention should therefore be given to study landslide susceptibility on the local and regional level.  For the Rwenzori Mountains, the

combination of regional landslide susceptibility with population density data allows to visualize areas where landslide risk could be particularly high and where research and policy-oriented action needs to be taken.

**Data Availability**

Landslide susceptibility maps presented in this study and landslide data used for their construction can be requested by contacting the corresponding author.

**Appendix A: InSAR processing to construct the TanDEM-X DEM**

Over the last two decades, Interferometric Synthetic Aperture Radar (InSAR) has been one of the main satellite-based tool used to evaluate ground displacements (Massonet & Feigl, 1998; Bürgmann et al., 2000; Hannsen et al., 2001). This technique also allows to construct DEMs and in particular the DLR (Deutsches Zentrum für Luft- und Raumfahrt - German Aerospace Center) TanDEM-X mission was specifically designed to generate a consistent global DEM (Deo et al., 2013). Indeed, within

the TanDEM-X bi-static mode (Moreira et al., 2004), two satellites in close formation acquire the radar complex images (amplitude + phase components) of the same area at the same time. Each pixel of a SAR image is represented by a complex number: the amplitude corresponds to the backscattered energy by the surface illuminated by the radar microwave impulse while the phase equals a fraction of the complete wavelength (having a value between 0 and $2\pi$). The phase difference from two SAR images (the interferogram) reveals variations in the distance between the ground and the satellites and appear as

coloured fringes. Since the TanDEM-X images are acquired simultaneously no ground deformation and atmospheric changes





will be observed between the two acquisitions and the interferogram contains only the topography phase component and random noise (Bürgmann et al., 2000; Hanssen, 2001). For this work, each pair of the TanDEM-X SAR images are processed with the ENVI-SARscape© software and we followed these steps to produce the DEMs:

(1) the images are co-registered (i.e. geometrically overlapped) using the amplitude components

(2) the phase difference $\Delta\varphi$ (the interferogram) is evaluated and contain just two components:

$$\Delta\varphi = \varphi_t + \varphi_n \qquad (A1)$$

where $\varphi_t$ is the topographic phase component (i.e. the DEM), $\varphi_n$ is the noise phase component. For the i-th pixel, $\Delta\phi_i$ ranges from 0 and $2\pi$, by looking at the $\Delta\phi$ of neighbour pixels and counting the fringe it is possible to evaluate the altitude. Indeed, each fringe corresponds to a variation on the altitude of the same value:

$$h_a = |\frac{\lambda R sin(\theta)}{2B_\perp}| \qquad (A2)$$

Where $h_a$ is called altitude of ambiguity (Massonet and Rabaute, 1993; Hanssen, 2001), $\lambda$ is the wavelength of the satellite (~3 cm for TanDEM-X), R is the satellite altitude and $B_\perp$ is the perpendicular baseline between the two satellites.

(3) To reduce the noise the image is multi-looked, i.e. the average over two pixels of SAR complex data is made in range (x)
and azimuth (y). As the initial Single Look Complex data have spatial resolutions of ~2.5 m x 2.5 m, the resulting DEM is obtained at a resolution of ~5 m.

(4) The SRTM-1arcsec DEM (Farr et al., 2000) is used to produce a synthetic interferogram, i.e. it is projected in radar looking geometry and then the altitude is converted in phase value. The synthetic interferogram is removed from the initial one to obtain the flattened interferogram. With TanDEM-X bi-static images, this difference represents the
topographic changes compared to the SRTM DEM.

(5) To further reduce the noise, an adaptive filter (Goldstein et al., 1998) is applied on the flattened interferogram to obtain the final filtered interferogram. The filter strength is evaluated for each pixel on the basis of the coherence (γ) value: the lower coherence, the stronger the filter. The coherence is the cross-correlation value between the phase of one pixel and its neighbours within a window with preselected dimensions (in azimuth and range) and it ranges from 0 (not coherent)
to 1 (complete coherence). This filter allows to smooth the phase of the noisiest pixels since their coherence is low.

(6) To convert the $\Delta\phi$ values of the filtered interferogram in elevation data the unwrapping step is required. Here we used the minimum cost flow algorithm (Goldstein et al., 1988; Costantini, 1998). We used a 0.25 unwrapping threshold (all the pixels with coherence lower than this value are discarded).

(7) The absolute calibrated and unwrapped phase is finally re-combined with the synthetic phase associated to the SRTM
DEM and it is converted to height. Then, the height map is geocoded using the SRTM DEM into the Lat/Lon cartographic system (WGS-84 ellipsoid).

The vertical precision of the measurement, $\sigma_z$, is a function of the acquisition geometry and the standard deviation of the phase, $\sigma_\phi$;



$$\sigma_z = (h_a/2 \pi)^* \sigma_\phi \qquad (A3)$$

where $h_a$ is the altitude of ambiguity (Eq.2). The standard deviation of the phase corresponds to the noise level of the interferogram and can be characterized, at first order, by the coherence parameter $\gamma$. For the Rwenzori InSAR DEMs produced from each image, the vertical error is on average < 7m.

Here we used 9 and 7 TanDEM-X couples in ascending and descending orbits respectively to construct two different DEMs that cover the entire Rwenzori region. We compared this DEMs with the SRTM-1 arc-sec and we noticed a horizontal shift that depends on the satellite acquisition geometry and it is principally due to the geocoding process, to the error on the orbital data and to the horizontal error on the SRTM (~20 m – Farr et al., 2000). Therefore, in order to limit this propagation, a mosaic of ascending and descending DEMs is constructed to cover the entire Rwenzori region. For the inhabited region, the vertical
error of this mosaic is on average <3m.

**Competing interests**

The authors declare that they have no conflict of interest.

**Acknowledgements**

This study was supported by the Belgium Science Policy (BELSPO) through the AfReSlide project BR/121/A2/AfReSlide in
the BRAIN program entitled 'Landslides in Equatorial Africa: Identifying culturally, technically and economically feasible resilience strategies', the RESIST project SR/00/3052 in the Stereo III program: REmote Sensing and In Situ detection and Tracking of geohazards and through the Vi-X project SR/00/150, in the Stereo II program (INTER Program, co-supported by FNR Luxembourg; Grant NTI_INSA0525, DLR). Support was also received from the Research Foundation Flanders (FWO) for a long research stay abroad. Additional support was received from the VLIR South-Initiative projects ZEIN2013Z145
entitled 'Diagnosis of land degradation processes, their socio-economical and physical controls and implications in the Mt Rwenzori region' and UG2017SIN208A105 entitled 'Enhancing community-based natural resources and hazard management in Rwenzori Mountains'. W.T. was supported by an ETH Zurich postdoctoral fellowship (Fel-45 15-1) and acknowledges the Uniscientia Foundation and the ETH Zurich Foundation.

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





## Figures

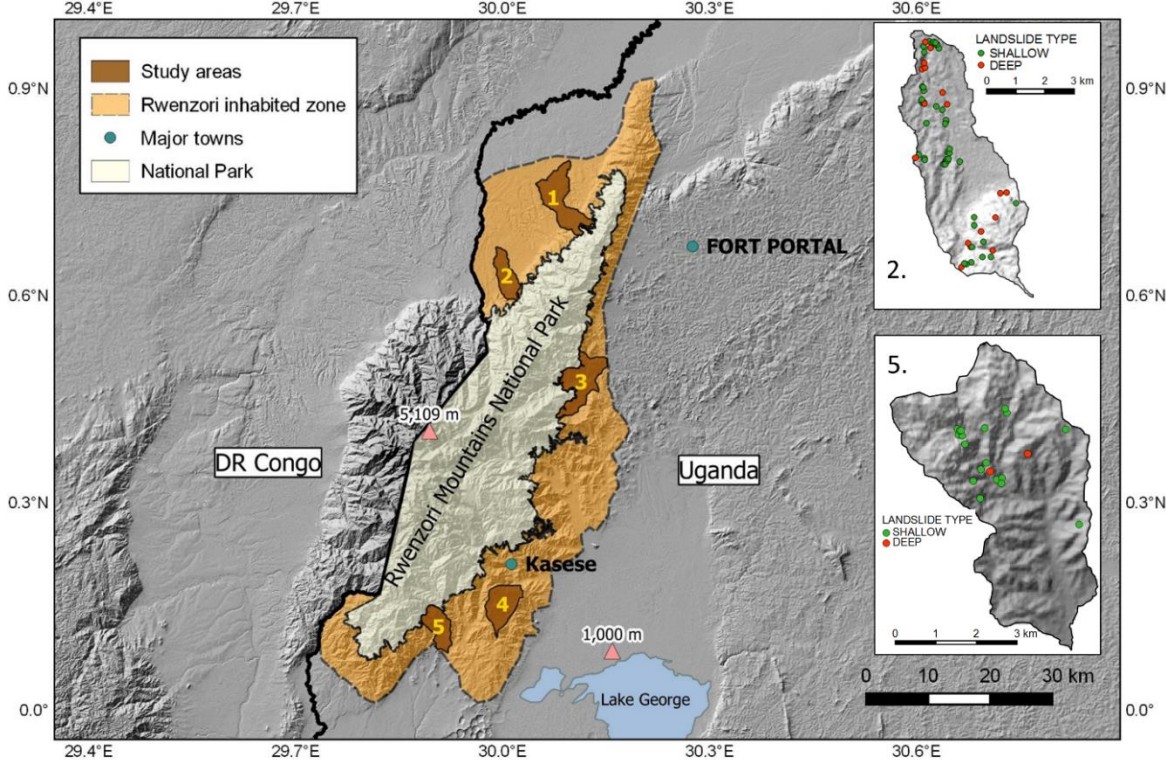

**Figure 1: Overview of the surveyed study areas on the Rwenzori Footslopes. The study areas are numbered from North to South: 1: Bundibugyo, 2: Nyahuka, 3: Kabonero, 4: Mahango and 5. Kyondo. For Nyahuka (2) and Kyondo (5) a detailed map is provided showing the locations of shallow and deep-seated slides. Landslide maps of other study areas can be found in Jacobs et al. (2016b).**





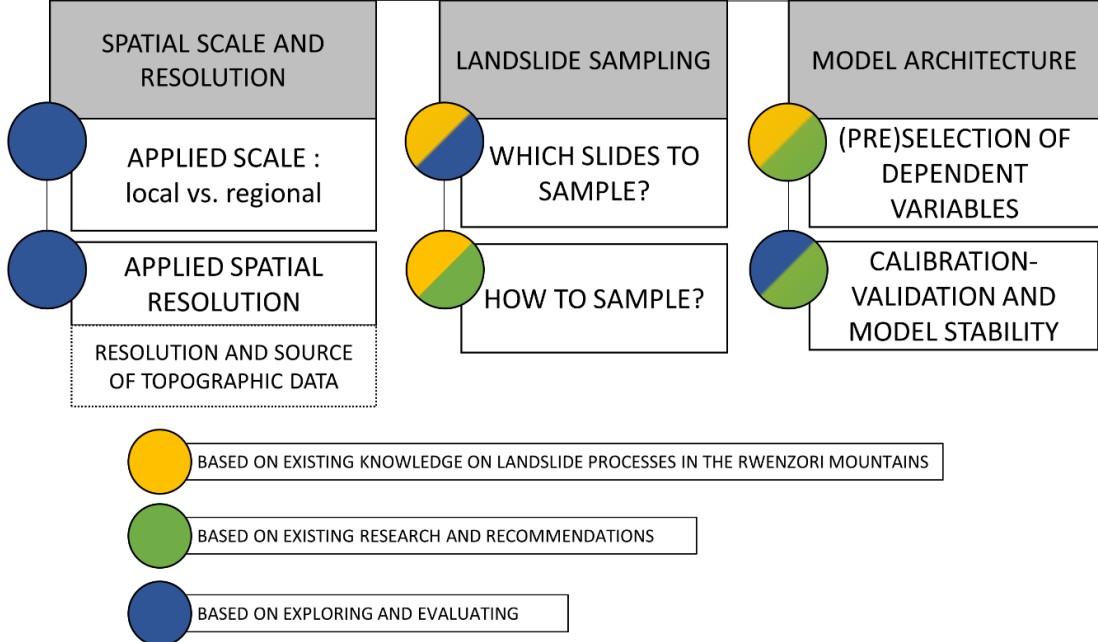

Figure 2: Schematic representation of the methodological options considered in this study and how they are evaluated and selected.





**Figure 3: a to e: Sensitivity (red), specificity (green) and prediction rate curve (blue) of the selected model variants. Bold line represents the means and shaded areas the standard deviation of the metric over 20 simulations. f: mean prediction rates for selected models on the regional and local levels.**





**Figure 4: Pair-wise comparison of the regional susceptibility model (left) and the local model (right) applied to the five case study areas. Each of the models are achieved by averaging results of the 20 model runs for the selected variant. *Indicates the average performance in AUC_{ROC} for the regional model as applied to the local study area.**




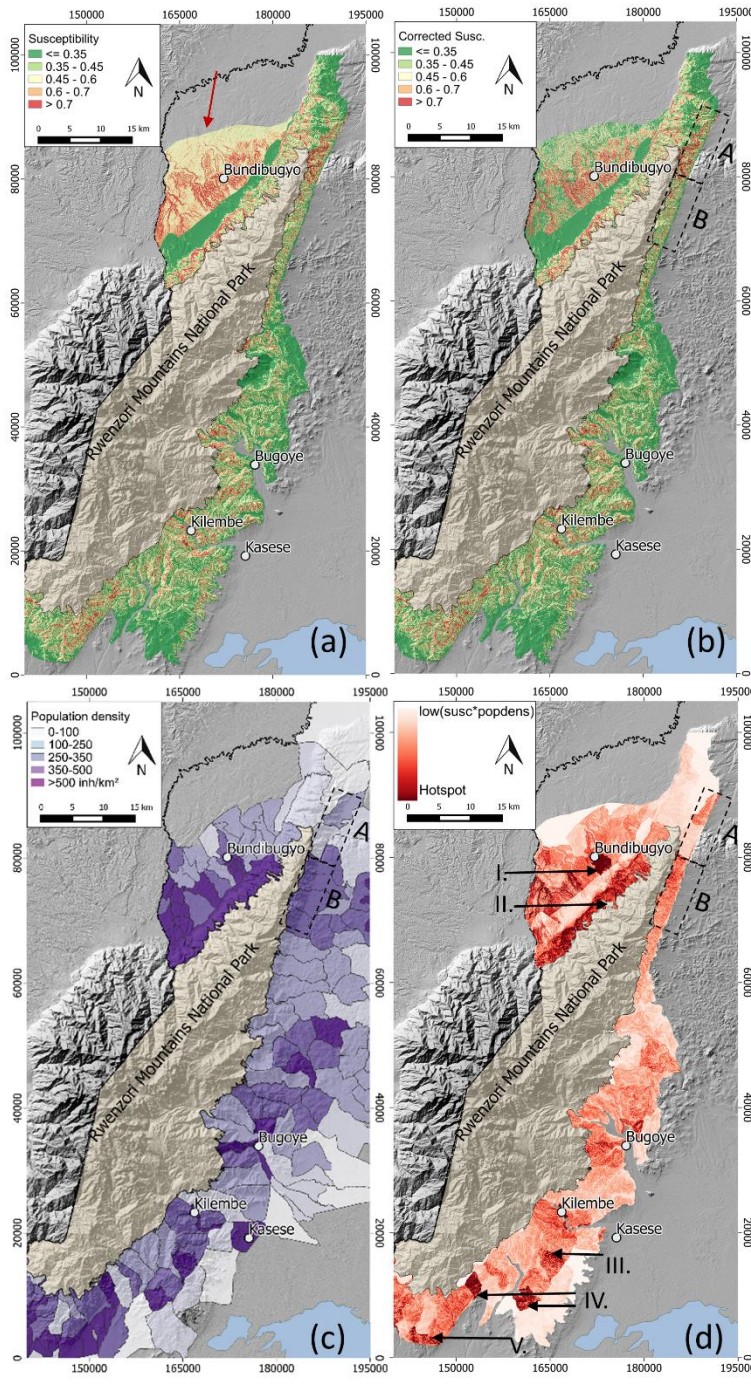

**Figure 5: Landslide susceptibility (LS susc) and population density (pop dens) in the Rwenzori Mountains inhabited zone. a: landslide susceptibility (red arrow indicates artefacts in regional landslide susceptibility model), b: corrected landslide susceptibility, c: population density at parish level (source: UBOS, 2003), d: preliminary identification of landslide risk hotspots. A and B indicate polygons discussed in the text. I to V indicate particular hotspot locations elaborated in the text.**





**Tables**

**Table 1: Overview of surveyed study areas (Fig.1) , their surface area and number of total, shallow (SLS) and deep-seated landslides (DSLS) mapped together with the prevailing lithologies in those study areas according to the GTK consortium (2012) and the average annual precipitation as simulated by Thiery et al. (2015).**

| Study area | Area (km²) | Elevation range (m a.s.l.) | Average slope (°) | # of SLS | # of DSLS | Total # of slides | Lithology | Average annual precipitation (mm) |
|---|---|---|---|---|---|---|---|---|
| Bundibugyo | 42.6 | 715-2,200 | 12 | 83 | 125 | 208 | Rift alluvium, Mica schists, Gneiss | 1,010 |
| Nyahuka | 20.4 | 830-2,200 | 12 | 48 | 17 | 65 | Rift Alluvium, Mica schists, Amphibolite, Gneiss | 1,540 |
| Kabonero | 39.8 | 1,400-2,300 | 20 | 53 | 17 | 70 | Gneiss, Amphibolite, Mica schists | 810 |
| Mahango | 29.6 | 1,240-2,200 | 20 | 69 | 22 | 91 | Gneiss | 930 |
| Kyondo | 20.4 | 1,130-2,140 | 20 | 18 | 2 | 20 | Gneiss, (Sericite) quartzite, Mica schists | 1,040 |
| Total | **153** | | | 271 | 183 | 454 | | |

**Table 2: Overview of the mean and standard deviation (S.D.) of AUCroc over 20 runs for the different model variants at each level. Selected model variants are marked in bold, values in italic indicate non-normally distributed samples (Shapiro-Wilk test, p<0.05). The null-hypothesis of homoscedasticity could only be rejected for the comparison of all variants in Nyahuka (p<0.05). Wherever applied, all non-parametric tests and the Welch-test confirm the results of the parametric tests and results are therefore combined**
10 **here. Symbols indicate significant differences detected between variants.**

| | | TANDEMX10 | TANDEMX20 | TANDEMX30 | ASTER30 | SRTM30 | SRTM90 |
|---|---|---|---|---|---|---|---|
| REGIONAL | Mean AUC$_{roc}$ | 0.71 | 0.71 | 0.71 | 0.69 | **0.71** | 0.68 |
| ◊ / ○ / Δ | S.D. AUC$_{roc}$ | 0.01 | 0.01 | 0.01 | 0.01 | **0.01** | 0.02 |
| BUNDIBUGYO | Mean AUCroc | 0.70 | 0.69 | 0.70 | 0.63 | **0.70** | 0.65 |
| ◊ / ○ / Δ | S.D. AUC$_{roc}$ | 0.03 | 0.02 | 0.01 | 0.02 | **0.03** | 0.02 |
| NYHUKA | Mean AUCroc | 0.69 | 0.69 | 0.64 | 0.67 | *0.68* | **0.74** |
| ◊ / □ / Δ | S.D. AUC$_{roc}$ | 0.04 | 0.03 | 0.05 | 0.07 | *0.04* | **0.06** |
| KABONERO | Mean AUCroc | 0.75 | **0.78** | 0.76 | 0.73 | *0.75* | 0.70 |
| ◊ / □ / ○ / Δ | S.D. AUC$_{roc}$ | 0.05 | **0.04** | 0.03 | 0.03 | *0.05* | 0.05 |
| KYONDO /MAHANGO | Mean AUCroc | *0.71* | **0.73** | 0.71 | *0.68* | 0.72 | 0.67 |
| ◊ / ○ / Δ | S.D. AUC$_{roc}$ | *0.04* | **0.04** | 0.03 | *0.05* | 0.03 | 0.03 |

Significant difference *between all variant*s (◊ at p<0.05)
Significant difference *between TANDEMX variant*s (□ at p<0.05)
Significant difference *between all variants at 30m* (○ at p<0.05)
Significant difference *between SRTM variants* ( Δ at p<0.05)

**Table 3: Overview of the frequency a variable is selected with a positive coefficient (+β, shaded cells) or negative coefficient (-β, unshaded cells) based on the AIC-criterion and the frequency that variable is found to be significant at a p<0.05 level for all local levels and the regional level. NP stands for Not Present in the study area, CURV stands for curvature**





| Variables | REGIONAL | | | | BUNDIBUGYO | | | | NYAHUKA | | | | KABONERO | | | | KYONDO/MAHANGO | | | |
|---|---|---|---|---|---|---|---|---|---|---|---|---|---|---|---|---|---|---|---|---|
| | +β | p<0.05 | -β | p<0.05 | +β | p<0.05 | -β | p<0.05 | +β | p<0.05 | -β | p<0.05 | +β | p<0.05 | -β | p<0.05 | +β | p<0.05 | -β | p<0.05 |
| RIFT ALLUVIUM | 20 | 20 | 0 | 0 | 19 | 15 | 1 | 0 | 19 | 11 | 1 | 0 | NP | NP | NP | NP | NP | NP | NP | NP |
| AMPHIBOLITE | 0 | 0 | 20 | 12 | NP | NP | NP | NP | 16 | 2 | 4 | 0 | 0 | 0 | 20 | 8 | NP | NP | NP | NP |
| SCHISTS | 6 | 0 | 14 | 0 | 20 | 9 | 0 | 0 | 19 | 5 | 1 | 0 | 0 | 0 | 20 | 1 | 0 | 0 | 20 | 14 |
| QUARTS | 16 | 0 | 4 | 0 | NP | NP | NP | NP | NP | NP | NP | NP | NP | NP | NP | NP | 18 | 1 | 2 | 0 |
| ELEVATION | / | / | / | / | 0 | 0 | 5 | 3 | 3 | 1 | 0 | 0 | 0 | 0 | 3 | 1 | 11 | 9 | 0 | 0 |
| SLOPE | 20 | 20 | 0 | 0 | 20 | 20 | 0 | 0 | 19 | 18 | 0 | 0 | 18 | 15 | 0 | 0 | 20 | 20 | 0 | 0 |
| TANG CURV | 0 | 0 | 15 | 13 | 1 | 1 | 3 | 1 | 0 | 0 | 3 | 2 | 0 | 0 | 15 | 13 | 0 | 0 | 7 | 3 |
| PROF CURV | 0 | 0 | 2 | 1 | 11 | 5 | 0 | 0 | 1 | 0 | 9 | 5 | 12 | 6 | 0 | 0 | 4 | 2 | 0 | 0 |
| TWI | 3 | 1 | 0 | 0 | 1 | 0 | 0 | 0 | 17 | 15 | 0 | 0 | 1 | 1 | 6 | 2 | 12 | 8 | 0 | 0 |
| N-S aspect | 1 | 1 | 1 | 0 | 0 | 0 | 5 | 3 | 2 | 1 | 3 | 1 | 2 | 1 | 0 | 0 | 2 | 2 | 3 | 1 |
| E-W aspect | 0 | 0 | 2 | 2 | 1 | 0 | 2 | 2 | 2 | 0 | 2 | 1 | 1 | 1 | 1 | 0 | 2 | 2 | 3 | 1 |
| NW-SE aspect | 2 | 0 | 1 | 1 | 2 | 1 | 6 | 5 | 1 | 0 | 4 | 2 | 1 | 0 | 2 | 0 | 3 | 0 | 0 | 0 |
| NE-SW aspect | 0 | 0 | 1 | 1 | 3 | 3 | 0 | 0 | 3 | 2 | 2 | 1 | 5 | 1 | 0 | 0 | 1 | 0 | 3 | 2 |
| PRECIPITATION | 11 | 8 | 0 | 0 | NP | NP | NP | NP | NP | NP | NP | NP | NP | NP | NP | NP | NP | NP | NP | NP |



**Table 4 Overview of the frequency a variable is selected with a positive coefficient (+β, shaded cells) or negative coefficient (-β, unshaded cells) based on the AIC-criterion and the frequency that variable is found to be significant on a p<0.05 level. This is given for the models on the regional level including all landslides, including only the shallow landslides and only the deep-seated landslides. The distinction between the latter two is made based on a 3m threshold of the depth of the sliding plane. NP stands for Not Provided as input to the model.**

| Variables | REGIONAL +β | p<0.05 | -β | p<0.05 | SHALLOW SLIDES REGIONAL +β | p<0.05 | -β | p<0.05 | DEEP-SEATED SLIDES REGIONAL +β | p<0.05 | -β | p<0.05 |
|---|---|---|---|---|---|---|---|---|---|---|---|---|
| RIFT ALLUVIUM | 20 | 20 | 0 | 0 | 18 | 6 | 2 | 0 | 20 | 20 | 0 | 0 |
| AMPHIBOLITE | 0 | 0 | 20 | 12 | 0 | 0 | 20 | 10 | 5 | 0 | 15 | 1 |
| SCHISTS | 6 | 0 | 14 | 0 | 8 | 0 | 12 | 1 | 14 | 0 | 6 | 0 |
| QUARTS | 16 | 0 | 4 | 0 | 15 | 0 | 5 | 0 | 3 | 0 | 10 | 0 |
| SLOPE | 20 | 20 | 0 | 0 | 20 | 20 | 0 | 0 | 20 | 20 | 0 | 0 |
| TANG CURV | 0 | 0 | 15 | 13 | 0 | 0 | 12 | 8 | 0 | 0 | 7 | 2 |
| PROF CURV | 0 | 0 | 2 | 1 | 0 | 0 | 1 | 0 | 2 | 1 | 0 | 0 |
| TWI | 3 | 1 | 0 | 0 | 6 | 4 | 0 | 0 | 1 | 1 | 1 | 1 |
| N-S (aspect) | 1 | 1 | 1 | 0 | 3 | 2 | 1 | 0 | 2 | 0 | 0 | 0 |
| E-W (aspect) | 0 | 0 | 2 | 2 | 1 | 1 | 0 | 0 | 0 | 0 | 1 | 0 |
| NW-SE (aspect) | 2 | 0 | 1 | 1 | 0 | 0 | 2 | 1 | 2 | 1 | 0 | 0 |
| NE-SW (aspect) | 0 | 0 | 1 | 1 | 0 | 0 | 3 | 3 | 1 | 0 | 1 | 1 |
| PRECIPITATION | 11 | 8 | 0 | 0 | 16 | 11 | 0 | 0 | 3 | 0 | 1 | 1 |
| TREE COVER | NP | NP | NP | NP | 1 | 0 | 2 | 0 | NP | NP | NP | NP |