# Peer review of "Field-based landslide susceptibility assessment in a data-scarce environment: the populated areas of the Rwenzori Mountains"

_Natural Hazards and Earth System Sciences, 2017_

## Referee Comment (RC1) · M. Jurchescu (Referee) · 6 Sep 2017

"General comments"

The paper brings about an interesting discussion on a subject that landslide literature could consider as being exhausted, namely on the landslide susceptibility assessment. The study successfully responds to two major objectives, as clearly stated in the Introduction section: i) providing new and important information for understanding regional-scale landslide susceptibility patterns and its conditioning factors in a landslide-threatened area of equatorial Africa (Rwenzori Mountains' Footslopes, Uganda), characterized by very limited data and hence necessitating extensive field in-

vestigations, and ii) achieving solid statistical modeling results by thoroughly discussing several methodological choices (scale, topographic data source, spatial resolution, selecting landslide type, landslide sampling, variable selection, deciding on calibration and validation data) and grounding these on previous theoretical knowledge or on running simulations and exploring their effects. The authors propose to address landslide susceptibility in the context of a data-scarce environment by: i) constructing a regional-scale model in two phases - calibration and validation using landslide inventories in representative case-studies, followed by the extrapolation to the rest of the area -, ii) deriving separate local-scale maps for the individual case-studies and iii) using the regional susceptibility map in combination with population density for a preliminary identification of landslide risk hotspots. Given the in-depth statistical analyses, the obtained maps can be considered valuable tools for directing and implementing policy actions in the region. Results are based on several model variants (varying according to the incorporated topographic data of different sources and spatial resolutions) being run for each scale (local / regional) and subsequently evaluated. Evaluations of obtained models (e.g. in terms of performance or variables' selection) as well as the identification of the optimal ones are thoroughly conducted, being based upon statistical significance tests. Additionally, differences induced by random landslide sampling are taken into account by running 20 simulations per variant. The manuscript is well structured and written and illustrations and tables are all necessary. In my opinion, the paper is worth publishing with only minor revisions.

"Specific comments"

1. One of the qualities of the paper consists in testing the effects of several methodological choices on model performances and, especially, in the thorough comparison of these model performances in order to allow for the selection of the appropriate model variants for each level. It is highlighted that tests used to assess statistical significance of differences in performance (in terms of AUC values) highly depend on the assumptions of normality and homogeneity of variance (homoscedasticity) of the examined

values. Several steps are therefore involved: i) distributions are tested for normality, ii) homoscedasticity is tested, by also using tests which are robust to large departures from normality (Fligner-Killeen test); iii) only then significance tests are run for assessing performance differences, varying in type according to their suitability to the different categories issued.

2. An important insight is brought with respect to the scale issue, a fundamental concept in many sciences, through comparisons among regional and local models. Specifically, it is shown, not only that the regional model, when applied (validated) at the local scale, is generally outperformed by the local models, but also that it lowers its own performance when interpreted in the context of the larger scale (from 0.71 to 0.65-0.70, with one exception, according to Fig. 4). I suggest testing if this lowering from regional to local level is statistically significant and recommend that these aspects be also mentioned in the respective paragraphs (section 3.3. page 9 lines 8-13 and section 4.1. page 11 lines 10-21). This would contribute to reminding us of the scale-dependency of model outcomes, i.e. that model results derived at one scale should not be transferred at another scale, but considered valid and interpreted in the specific context of the scale they were obtained at.

3. I notice the description of data and methodology used is particularly accurate to allow reproduction. This is enhanced by the appendix describing the processing of TanDEM-X images to construct the respective DEMs as well as by the explicit Data Availability expression at the end of the paper.

4. In my opinion, other previous similar methodological questions on the impacts of varying data (of different sources, resolutions and accuracies: e.g. raised by Lee et al. 2004, Catani et al. 2013, Fressard et al. 2014) or of various modeling settings (e.g. by Poli & Sterlacchini 2007, Yilmaz 2010, Hussin et al. 2016 for landslide sampling strategies; by Heckmann et al. 2014, Petschko et al. 2014 for subdivision in calibration and validation datasets) on model performances could complement the ones already mentioned in the Methodology section (e.g. section 3, page 4 lines 15-17; section 3.1.
page 4 line 32 – page 5 line 1, sections 3.2.1, 3.2.4.). Recommended would also be the use of "e.g.", since citations here cannot be exhaustive.

5. Please also refer to motivations brought by previous papers in making methodological decisions. For example, in section 3.2.1. Landslide sampling, page 6 lines 12-13, I suggest citing Van Den Eeckhaut et al. (2006) since these authors give the same motivation for selecting only the central cells of landslide depletion zones (i.e. avoiding spatial autocorrelation and the violation of the assumption of data statistic independence).

6. If and where possible, please also briefly complement the references to outcomes of similar experiments conducted on the effect of using different data sources and spatial resolutions (e.g. Catani et al. 2013, Fressard et al. 2014) when discussing the results achieved in the current paper (section 5.1).

7. Section 3.2.1., page 6 line 6-7: I would not agree that the sentence listing landslide sampling techniques also includes: "or construct a buffer zone around (portions of) the landslide to represent the conditions under which the landslide occurred (Dai and Lee, 2003; Suzen and Doyuran, 2004; Van den Eeckhaut et al., 2006; Che et al., 2012; Hussin et al., 2016)". To my knowledge, buffer zones around landslides are drawn in order to eliminate those areas from sampling absences and not presences of landslides. Please check this and reword the phrase accordingly.

8. Figure 2 nicely presents the various methodological aspects considered in the research as well as the manner in which they were evaluated. However, it is not clear what does the "(Pre)Selection of Dependent Variables" refer to. Should there not be "(Pre)selection of Independent Variables" i.e. of explanatory variables (since the discussion on the selection of the dependent variable, i.e. landslide occurrences, is made under "Landslide sampling")?

9. I think the title of figure 4 should also include the explanation that small circles represent the mapped landslides.

10. I would suggest modifying the title of section 4.2. from "Separation of landslide types" into "Separation of landslide types and controlling variables", since this section presents results both in terms of the effect of distinguishing landslide categories and in terms of the significant explanatory variables changing according to the landslide type. This would also enable a more obvious linkage between the Results and the Discussions sections.

"Technical corrections"

- Abstract, page 1 line 20: Please delete "the" from the sentence "Topographic data is extracted from different the digital elevation models (DEMs) based….";

- Section 2, page 3 line 19: I am not sure the word "inventorized" exists, I would suggest to replace it with "inventoried";

- same correction is suggested for the next line;

- Section 2, page 3 lines 20-21: I would suggest replacing the sentence "For these two study areas a detail of the landslide inventory is given in Fig. 1" with "For these two study areas, maps of the landslide inventory are given as details in Fig. 1";

- Section 2, page 3 line 25: Please delete the word "mostly" from "In Nyahuka however, mostly shallow landslides prevail";

- Section 2, page 3 line 25-26: Please reverse the order of words, i.e. "surveyed study areas" instead of "study areas surveyed";

- Section 2, page 3 line 28-29: Please insert the word "which" as follows: "In total this inventory contains 454 landslides which were used for the susceptibility modeling";

- Section 3, page 4 line 2: I would suggest using the plural, i.e. "landslide occurrences";

- Section 3, page 4 line 3: "as the dependent variable" instead of "as dependent variable";

- Section 3.2.3, page 7 line 26: Please insert the word "of" as follows: "strong arguments support the use of elevation as an explanatory variable";

- Section 5.3, page 14 line 22: I suggest using the plural, i.e. "their effects";

- Section 6, page 16 lines 13-14: Please correct the sentence as follows: "In parallel to smaller scale landslide susceptibility studies, adequate attention should therefore also be given to investigating landslide susceptibility on the local and regional levels."

---

## Referee Comment (RC2) · J. Hervás (Referee) · 23 Oct 2017

General comments: Paper summarized objectives, methodology and results are already well commented by the first reviewer, therefore I will skip this part in my comments. Likewise, the manuscript is well structured and written, and figures and tables are quite illustrative. The manuscript is considered valuable for publication with minor revision.

Specific comments: Most positive comments and suggestions by the first reviewer are also subscribed here. Additional comments follow. 1. In the paper title, the term "Field-based..." does not appear to properly reflect the contents of the paper. In addition,

field investigations are mainly dealt with in previous work by the main author. Therefore I suggest to change this somehow misleading title. 2. Page 2, lines 12-13: Do you mean reliable landslide susceptibility "models" or rather "data" or "assessments"? 3. Page 5, line 10: A reference should be provided for TanDEM-X DEM as done for the other DEMs. 4. Page 5, lines 6-18: Although four DEM resolutions are said to be used, the last sentence appears to indicate six resolutions. Please clarify. Also, adding a table would improve the readability of this part. 5. Page 9, lines 20-21: Even though the focus of this part in placed on risk hotspots not on risk assessment, the census data used (from 2002) appears quite outdated considering that a more recent census of 2014 provisionally reports an outstanding 43% increase of total population in the country in that period. 6. Page 12, line 27 – Page. 13, line 2: Can the findings from DEMs quality comparison be extrapolated to non-tropical regions, i.e. to areas where more cloud-free ASTER imagery can be acquired and thus better ASTER-based DEM model can be constructed (e.g. to Europe)? Please explain. 7. Page 15, line 24 (and page 9, line 23): Within-parish distribution of population densities could be somehow determined through mapping built-up areas using optical satellite imagery, thus providing a better identification of landslide risk hotspots when combined with landslide susceptibility levels. Please consider this in the revision. 8. Page 18, lines 8-9: It would be helpful to explain possible DEM mosaicking-derived issues (e.g. boundary effects) and the solutions implemented if any. 9. Page 26: In Figure 3 caption it would be helpful to refer to table 2. 10. Page 30: Table 3: Please clarify why there is no lithological variable called gneiss, despite it is reported in the text to be the reference lithology in the region. Additionally, do you mean by "quarts" "quartzites"? Please clarify or correct. 11. Page 31: Table 4: Same as previous comment.

Technical corrections: Page 1, line 19: I suggest to remove comma. Page 1, line 20: "the" must be removed. Page 1, line 24: Space is needed between 10 and m for consistency throughout the manuscript. Page 3 Line 28: I suggest to remove "depth". Page 4, line 16: Add comma after "al." Page 5, line 21: Add "s" to resolution. Page 7, line 1: The citation "Yilmaz et al., 2010" should be "Yilmaz, 2010". Page 7, line

19: The citation "Dai et al., 2003" should be "Dai and Lee, 2003". Page 7, line 26: Please insert "of" between "use" and "elevation". Page 8, line 2: I suggest to delete this sentence, as it is already more appropriately included two sentences afterwards. Page 8, line 26: The initial letter of tanDEM-X should be capitalized. Page 10, line 14: TanDEMX20 should be all capitalized according to the notation previously established by the authors. Page 10, line 19: Please pluralize "variant". Page 11, line 4: Please replace "this" with "these". Page 12, lines 21-25: Please insert some comma in this very long sentence. Page 16, line 22: Change Massonet to Massonnet. Also change "&" to "and". In addition, the citation "Hannsen et al., 2001" does not exit in references, Please correct spelling or references. Page 17, lines 7-8: Please use same symbols noting phase as before. Should they mean something else please explain them Page 17, line 12: Change Massonet to Massonnet. Page 17, lines 12-13: Please explain also the $\theta$ meaning. Page 17, line 26: Add a space before "values". Page 18, line 6: Change "this" to "these". Page 19, lines 8-9: Please provide EGU abstract number. Page 21, line 5: Change Switserland to Switzerland. Page 21, lines 19-21: Please complete reference. Page 22, lines 20-21: Please complete reference. Page 23, line 3: Please correct authors style.

---

## Author Comment (AC1) · 16 Nov 2017

Dear members of the editorial board,

Dear referees,

We thank you for providing us with the evaluation on our manuscript. With this reply we hope to provide adequate answers to the comments of the reviewers. This is done in a point-by-point fashion below. First the comment of the referee (RC) is given first, after which our response (AC) is given in normal font. We strongly appreciate the insight and feedback provided by both referees and are convinced these have helped

in further improving the manuscript. All changes made are indicated through a page and line number of a revised version of the manuscript that can be consulted in the supplement.

Response to the comments of the Referee 1: Dr. Marta-Cristina Jurchescu:

(1) General evaluation by referee

RC: The paper brings about an interesting discussion on a subject that landslide literature could consider as being exhausted, namely on the landslide susceptibility assessment. The study successfully responds to two major objectives, as clearly stated in the Introduction section: i) providing new and important information for understanding regional-scale landslide susceptibility patterns and its conditioning factors in a landslide-threatened area of equatorial Africa (Rwenzori Mountains' Footslopes, Uganda), characterized by very limited data and hence necessitating extensive field investigations, and ii) achieving solid statistical modeling results by thoroughly discussing several methodological choices (scale, topographic data source, spatial resolution, selecting landslide type, landslide sampling, variable selection, deciding on calibration and validation data) and grounding these on previous theoretical knowledge or on running simulations and exploring their effects. The authors propose to address landslide susceptibility in the context of a data-scarce environment by: i) constructing a regional scale model in two phases - calibration and validation using landslide inventories in representative case-studies, followed by the extrapolation to the rest of the area -, ii) deriving separate local-scale maps for the individual case-studies and iii) using the regional susceptibility map in combination with population density for a preliminary identification of landslide risk hotspots. Given the in-depth statistical analyses, the obtained maps can be considered valuable tools for directing and implementing policy actions in the region. Results are based on several model variants (varying according to the incorporated topographic data of different sources and spatial resolutions) being run for each scale (local / regional) and subsequently evaluated. Evaluations of obtained models (e.g. in terms of performance or variables' selection) as well as the identification of

the optimal ones are thoroughly conducted, being based upon statistical significance tests. Additionally, differences induced by random landslide sampling are taken into account by running 20 simulations per variant. The manuscript is well structured and written and illustrations and tables are all necessary. In my opinion, the paper is worth publishing with only minor revisions.

AC: The authors appreciate the positive and in depth evaluation of the manuscript's content by referee #1 both with regard to the thematic content and the methodologies applied.

(2) Specific comments by the referee

1. RC: One of the qualities of the paper consists in testing the effects of several methodological choices on model performances and, especially, in the thorough comparison of these model performances in order to allow for the selection of the appropriate model variants for each level. It is highlighted that tests used to assess statistical significance of differences in performance (in terms of AUC values) highly depend on the assumptions of normality and homogeneity of variance (homoscedasticity) of the examined values. Several steps are therefore involved: i) distributions are tested for normality, ii) homoscedasticity is tested, by also using tests which are robust to large departures from normality (Fligner-Killeen test); iii) only then significance tests are run for assessing performance differences, varying in type according to their suitability to the different categories issued.

AC: The referee supports our evaluation of normality and homoscedasticity prior to implementing significance tests for the variants' performance evaluation.

2. RC: An important insight is brought with respect to the scale issue, a fundamental concept in many sciences, through comparisons among regional and local models. Specifically, it is shown, not only that the regional model, when applied (validated) at the local scale, is generally outperformed by the local models, but also that it lowers its own performance when interpreted in the context of the larger scale (from 0.71 to

0.65-0.70, with one exception, according to Fig. 4). I suggest testing if this lowering from regional to local level is statistically significant and recommend that these aspects be also mentioned in the respective paragraphs (section 3.3. page 9 lines 8-13 and section 4.1. page 11 lines 10-21). This would contribute to reminding us of the scale-dependency of model outcomes, i.e. that model results derived at one scale should not be transferred at another scale, but considered valid and interpreted in the specific context of the scale they were obtained at.

AC: The referee suggests to further improve the discussion related to the applied scale by providing additional testing of the performance decrease of regional models tested on the local scale. We therefore now compare the regional models performance when applied to the individual local scales to the regional model performance when validated on the regional level. On two out of four local levels this decrease in performance is significant indicating that models calibrated and validated at a certain scale cannot readily be interpreted at a larger scale. This is now added to the manuscript in the methodology, results and discussion section: on pag.9 lines 12-13, pag. 11 lines 19-24, pag. 14 lines 15-17.

3. RC: I notice the description of data and methodology used is particularly accurate to allow reproduction. This is enhanced by the appendix describing the processing of TanDEM-X images to construct the respective DEMs as well as by the explicit Data Availability expression at the end of the paper.

AC: We are happy to learn that the reviewer appreciates the description of the TanDEM-X DEM production in the appendix.

4. RC: In my opinion, other previous similar methodological questions on the impacts of varying data (of different sources, resolutions and accuracies: e.g. raised by Lee et al. 2004, Catani et al. 2013, Fressard et al. 2014) or of various modeling settings (e.g. by Poli & Sterlacchini 2007, Yilmaz 2010, Hussin et al. 2016 for landslide sampling strategies; by Heckmann et al. 2014, Petschko et al. 2014 for subdivision in calibration

and validation datasets) on model performances could complement the ones already mentioned in the Methodology section (e.g. section 3, page 4 lines 15-17; section 3.1. page 4 line 32 – page 5 line 1, sections 3.2.1, 3.2.4.). Recommended would also be the use of "e.g.", since citations here cannot be exhaustive.

AC: We have complemented the references in the manuscript by suggestions made by the referee in order to strengthen the methodological section. We have added references in section 3, e.g. pag. 4 lines 15-17, pag 5 line 4-5. In order to keep the text concise we have, as suggested by the referee, used 'e.g.' wherever other references could be added and we were not exhaustive.

5. RC: Please also refer to motivations brought by previous papers in making methodological decisions. For example, in section 3.2.1. Landslide sampling, page 6 lines 12-13, I suggest citing Van Den Eeckhaut et al. (2006) since these authors give the same motivation for selecting only the central cells of landslide depletion zones (i.e. avoiding spatial autocorrelation and the violation of the assumption of data statistic independence).

AC: This reference is now taken into account: pag. 6 line 13.

6. RC: If and where possible, please also briefly complement the references to outcomes of similar experiments conducted on the effect of using different data sources and spatial resolutions (e.g. Catani et al. 2013, Fressard et al. 2014) when discussing the results achieved in the current paper (section 5.1).

AC: This is taken into account: pag. 12 line 29.

7. RC: Section 3.2.1., page 6 line 6-7: I would not agree that the sentence listing landslide sampling techniques also includes: "or construct a buffer zone around (portions of) the landslide to represent the conditions under which the landslide occurred (Dai and Lee, 2003; Suzen and Doyuran, 2004; Van den Eeckhaut et al., 2006; Che et al., 2012; Hussin et al., 2016)". To my knowledge, buffer zones around landslides are

drawn in order to eliminate those areas from sampling absences and not presences of landslides. Please check this and reword the phrase accordingly.

AC: A buffer area can indeed be used either to exclude those areas from the sampling of non-landslide points, but are likewise used to indicate areas of undisturbed but similar conditions under which the landslide occurs. The latter is often referred to as the 'seed cell' method (Suzen and Doyuran, 2004; Che et al., 2012). This clarification is made in the revised version of the manuscript: pag. 6 line 6.

8. RC: Figure 2 nicely presents the various methodological aspects considered in the research as well as the manner in which they were evaluated. However, it is not clear what does the "(Pre)Selection of Dependent Variables" refer to. Should there not be "(Pre)selection of Independent Variables" i.e. of explanatory variables (since the discussion on the selection of the dependent variable, i.e. landslide occurrences, is made under "Landslide sampling")?

AC: The referee is correct in pointing out this mistake: Fig. 2 is corrected

9. RC: I think the title of figure 4 should also include the explanation that small circles represent the mapped landslides.

AC: This suggestion is now included in the revised version of the manuscript: figure caption of Fig. 4

10. RC: I would suggest modifying the title of section 4.2. from "Separation of landslide types" into "Separation of landslide types and controlling variables", since this section presents results both in terms of the effect of distinguishing landslide categories and in terms of the significant explanatory variables changing according to the landslide type. This would also enable a more obvious linkage between the Results and the Discussions sections.

AC: It indeed will improve the clarity of the results section if the title of section 4.2 is modified. Similar to the first title of the result section 'Influence of model's spatial

resolution, topographic data source and scale', we modified the consecutive title from 'Separation of landslide types' to 'Influence of the separation of landslide types' as to present the results in a straightforward, uniform manner: pag. 11 line 27

(3) Technical comments by the referee AC: The technical corrections suggested by the referee are taken into account in the new version of the manuscript (in track changes).

Other changes to the manuscript: In the previous version of the manuscript we used a SRTM and ASTER DEM that resulted from the resampling to precisely $30*30m^2$ resolution based on a warped product in UTM coordinates that carried the original resolution (ca 30.7*30.8 $m^2$ resolution at the equator). We have now improved this by combining the warping from the WGS product to UTM coordinates and resampling to precisely $30*30m^2$ resolution in one step. This in principle entails less manipulation of the original data but de facto results in only minor changes in the final products used. With regard to the manuscript, this adjustment as a result does not imply significant changes to the findings with the exception of a slightly more nuanced interpretation of the ASTER variants' performances, which are now significantly outperformed by InSAR alternatives on 3 levels (compared to 4 levels in the previous version of the manuscript). For clarity, track changes have also been applied to these changes. Tables 2, 3 and 4 were updated as well as figures 3, 4 and 5. With regard of the slope threshold we use to correct our regional susceptibility model, we have decided to deploy a slightly more conservative cut-off value of 3° slope gradient in lieu of 5°. This change is reflected in minor adjustments to Figure 5 b and d. Finally, we've made a few clarifications in the manuscript, all of which are indicated in track changes in the document here in supplement.

Sincerely,

Liesbet Jacobs

On behalf of all the co-authors

[Figure]

Liesbet.jacobs@vub.be PhD researcher – VUB and RMCA AfReSlide project: http://afreslide.africamuseum.be/ Vrije Universiteit Brussel Pleinlaan 2, 1050 Brussels, Belgium +3226293556

Please also note the supplement to this comment:
https://www.nat-hazards-earth-syst-sci-discuss.net/nhess-2017-259/nhess-2017-259-AC1-supplement.pdf

**Supplement:**

[revised manuscript text omitted]

---

## Author Comment (AC2) · 16 Nov 2017

Dear members of the editorial board, Dear referees,

We thank you for providing us with the evaluation on our manuscript. With this reply we hope to provide adequate answers to the comments of the reviewers. This is done in a point-by-point fashion below. First the comment of the referee (RC) is given first, after which our response (AC) is given in normal font. We strongly appreciate the insight and feedback provided by both referees and are convinced these have helped in further improving the manuscript. All changes made are indicated through a page and line number of a revised version of the manuscript that can be consulted in the

supplement.

(1) General evaluation by referee

RC: General comments: Paper summarized objectives, methodology and results are already well commented by the first reviewer, therefore I will skip this part in my comments. Likewise, the manuscript is well structured and written, and figures and tables are quite illustrative. The manuscript is considered valuable for publication with minor revision.

AC: We are happy to learn that Referee #2 agrees with the positive evaluation by referee #1.

(2) Specific comments by the referee

1. RC: 1. In the paper title, the term "Fieldbased.. ." does not appear to properly reflect the contents of the paper. In addition field investigations are mainly dealt with in previous work by the main author. Therefore I suggest to change this somehow misleading title.

AC: It is correct that the manuscript contains limited information on the data collection itself as this is described in other publications (indicated in lines 20-21 pag. 3). However, we were asked in the initial submission stage to redirect the focus of the manuscript more towards the site application in a data-scarce environment. This includes a large component of field work, hence the choice for this title. The susceptibility analysis is in this sense field-based: the landslide data driving the analysis was collected mostly through field surveys.

2. RC: Page 2, lines 12-13: Do you mean reliable landslide susceptibility "models" or rather "data" or "assessments"?

AC: Indeed we mean "reliable susceptibility assessments": this is corrected in the revised manuscript: pag. 2 line 13.

[Figure]

3. RC: Page 5, line 10: A reference should be provided for TanDEM-X DEM as done for the other DEMs

AC: The reference 'Deo et al., 2013' is now included here: pag. 5 line 14.

4. RC: Page 5, lines 6-18: Although four DEM resolutions are said to be used, the last sentence appears to indicate six resolutions. Please clarify. Also, adding a table would improve the readability of this part.

AC: Indeed four different DEM resolutions are tested, but at 30 m resolution, three different DEMs are compared as described in that paragraph. This results in a total of six unique combinations of DEM type and resolution (pag. 5 lines 23-24). These combinations are described in lines 20-21 and thus an additional table seems of limited added value.

5. RC: Page 9, lines 20-21: Even though the focus of this part in placed on risk hotspots not on risk assessment, the census data used (from 2002) appears quite outdated considering that a more recent census of 2014 provisionally reports an outstanding 43% increase of total population in the country in that period.

AC: Indeed, the 2014 census data would be a better reflection of the current population density, however, the data on parish level has not been made available. Therefore, at this scale we only have the 2002 census data at our disposal.

6. RC: Page 12, line 27 – Page. 13, line 2: Can the findings from DEMs quality comparison be extrapolated to non-tropical regions, i.e. to areas where more cloud-free ASTER imagery can be acquired and thus better ASTER-based DEM model can be constructed (e.g. to Europe)? Please explain.

AC: In this manuscript, the suitability of different DEMs was evaluated for one region only. Therefore, a blind extrapolation of these findings to other regions should be avoided as DEM quality, regardless of the type of DEM, is inherently dependent on the quality of the acquired data used to construct these DEMs which can differ from

region to region. However, similar findings through direct comparison of e.g. the SRTM DEM and ASTER DEM are available in the literature (e.g. Guth, 2010 , Li et al., 2013 ). These references are now added to the manuscript: pag 13, lines 3-4.

7. RC: Page 15, line 24 (and page 9, line 23): Within-parish distribution of population densities could be somehow determined through mapping built-up areas using optical satellite imagery, thus providing a better identification of landslide risk hotspots when combined with landslide susceptibility levels. Please consider this in the revision.

AC: Population estimates based on optical satellite data in the tropics, although being a promising technique, is rare, especially at regional scales for rural areas. This requires very high resolution satellite data. Automatic mapping is complex in densely vegetated area with scattered houses. The perspective for a more detailed and reliable population density estimation is now included in the discussion: pag 15 lines 32-34.

8. RC: Page 18, lines 8-9: It would be helpful to explain possible DEM mosaicking-derived issues (e.g. boundary effects) and the solutions implemented if any.

AC: Indeed, seaming issues can arise when combining several images. However, mosaicking was performed while avoiding seams in the study areas used for the model calibration and validation. This is now specified in the aforementioned section: pag. 18 lines 22-23.

9. RC: Page 26: In Figure 3 caption it would be helpful to refer to table 2.

AC: This figure caption is now adjusted accordingly

10. RC: Page 30: Table 3: Please clarify why there is no lithological variable called gneiss, despite it is reported in the text to be the reference lithology in the region. Additionally, do you mean by "quarts" "quartzites"? Please clarify or correct.

AC: Gneiss is not mentioned in table 3 and 4 because it is the reference lithology: i.e. this lithology is coded by a 0-value combination of all other lithological dummy variables relevant to the investigated study area. If in one study area, n lithological classes

occur, among which gneiss, these n lithological classes will be coded by n-1 variables indicating the presence/absence of the n-1 lithological classes that are not gneiss. The presence of gneiss is then coded by a 0-value for all the n-1 other lithological classes. By including all n variables indicating the binary presence/absence of all lithological classes, including gneiss, a 'dummy trap' would occur, i.e. there is one redundant variable. As a result of these n-1 dummy transformations of the n lithological classes, the reference lithology does not have its own beta coefficient in the regression, i.e. the beta value, and the corresponding Odds Ratio (OR) should be interpreted relative to the reference value. Examples of such approaches can be found in Dai and Lee (2002) and Goverski et al. (2006) . These references are now added to the manuscript: pag; 7 line 6. A negative beta-coefficient for variables of the n-1 lithological classes indicate a OR below 1, i.e. a lower probability of landslides compared to the reference lithology and vice versa. The word "quarts" has been replace by the correct term "quartzites".

11. RC: Page 31: Table 4: Same as previous comment AC: Idem as 10.

(3) Technical comments by the referee The technical corrections suggested by the referee are taken into account in the new version of the manuscript (in track changes).

Other changes to the manuscript: In the previous version of the manuscript we used a SRTM and ASTER DEM that resulted from the resampling to precisely $30*30m^2$ resolution based on a warped product in UTM coordinates that carried the original resolution (ca $30.7*30.8$ $m^2$ resolution at the equator). We have now improved this by combining the warping from the WGS product to UTM coordinates and resampling to precisely $30*30m^2$ resolution in one step (Li and Goodchild, 2016 ). This in principle entails less manipulation of the original data but de facto results in only minor changes in the final products used. With regard to the manuscript, this adjustment as a result does not imply significant changes to the findings with the exception of a slightly more nuanced interpretation of the ASTER variants' performances, which are now significantly outperformed by InSAR alternatives on 3 levels (compared to 4 levels in the previous version of the manuscript). For clarity, track changes have also been applied to these

changes. Tables 2, 3 and 4 were updated as well as figures 3, 4 and 5. With regard of the slope threshold we use to correct our regional susceptibility model, we have decided to deploy a slightly more conservative cut-off value of $3°$ slope gradient in lieu of $5°$. This change is reflected in minor adjustments to Figure 5 b and d. Finally, we've made a few clarifications in the manuscript, all of which are indicated in track changes in the document here in supplement.

Sincerely,

Liesbet Jacobs

On behalf of all the co-authors

Liesbet.jacobs@vub.be PhD researcher – VUB and RMCA AfReSlide project: http://afreslide.africamuseum.be/ Vrije Universiteit Brussel Pleinlaan 2, 1050 Brussels, Belgium +3226293556

Please also note the supplement to this comment:
https://www.nat-hazards-earth-syst-sci-discuss.net/nhess-2017-259/nhess-2017-259-AC2-supplement.pdf

**Supplement:**

[revised manuscript text omitted]